

# State-space modeling of the dynamics of temporal plant cover using visually determined class data

Hiroki Itô

Hokkaido Research Center, Forestry and Forest Products Research Institute, Toyohira-ku, Sapporo, Japan

## ABSTRACT

A lot of vegetation-related data have been collected as an ordered plant cover class that can be determined visually. However, they are difficult to analyze numerically as they are in an ordinal scale and have uncertainty in their classification. Here, I constructed a state-space model to estimate unobserved plant cover proportions (ranging from zero to one) from such cover class data. The model assumed that the data were measured longitudinally, so that the autocorrelations in the time-series could be utilized to estimate the unobserved cover proportion. The model also assumed that the quadrats where the data were collected were arranged sequentially, so that the spatial autocorrelations also could be utilized to estimate the proportion. Assuming a beta distribution as the probability distribution of the cover proportion, the model was implemented with a regularized incomplete beta function, which is the cumulative density function of the beta distribution. A simulated dataset and real datasets, with one-dimensional spatial structure and longitudinal survey, were fit to the model, and the parameters were estimated using the Markov chain Monte Carlo method. Then, the validity was examined using posterior predictive checks. As a result of the fitting, the Markov chain successfully converged to the stationary distribution, and the posterior predictive checks did not show large discrepancies. For the simulated dataset, the estimated values were close to the values used for the data generation. The estimated values for the real datasets also seemed to be reasonable. These results suggest that the proposed state-space model was able to successfully estimate the unobserved cover proportion. The present model is applicable to similar types of plant cover class data, and has the possibility to be expanded, for example, to incorporate a two-dimensional spatial structure and/or zero-inflation.

# INTRODUCTION

There is a vast amount of historical data regarding plant abundance that were recorded as plant abundances in an ordered cover class, e.g., the Braun-Blanquet classification (*Podani, 2006*; *Irvine & Rodhouse, 2010*; *Damgaard, 2014*), much of which was determined visually. In many cases, such data are difficult to treat numerically; they are typically recorded in an "ordinal scale" so that standard arithmetic operations, such as addition or subtraction, are

Corresponding author
Hiroki Itô, abies.firma@gmail.com

not applicable (*Dale, 1989*; *Podani, 2006*). In addition, the uncertainty derived from the visual classification of such data tends to be ignored in analyses.

However, attempts to estimate unobserved "true" plant cover (the proportion in a unit area) from the ordered class data have been developed along with progress in statistical methods in the field of ecology (*Irvine & Rodhouse, 2010*; *Damgaard, 2014*; *Herpigny & Gosselin, 2015*; *Irvine, Rodhouse & Keren, 2016*; *Irvine et al., 2019*; *Damgaard & Irvine, 2019*). Ordered class data are typically modeled using ordered logit (cumulative logit) models, but the interpretation of the models has been known to be rather complicated (*Herpigny & Gosselin, 2015*; *Irvine, Rodhouse & Keren, 2016*).

However, some attempts have been made to model the plant cover proportion, assuming that this proportion follows the beta distribution (*Chen et al., 2008*; *Irvine & Rodhouse, 2010*; *Damgaard, 2014*; *Irvine, Rodhouse & Keren, 2016*; *Irvine et al., 2019*; *Damgaard & Irvine, 2019*). For example, *Damgaard (2014)* modeled the plant cover class as determined visually using the incomplete beta function based on the beta distributions of the plant cover. *Herpigny & Gosselin (2015)* incorporated zero-inflation, accounting for the excess zeros in the class data, into the model. *Irvine et al. (2019)* have proposed a Bayesian hierarchical framework accounting for true and false zeros in the class data as well as misclassification of the classes. *Damgaard & Irvine (2019)* comprehensively discussed this subject.

In recent decades, state-space models have been applied to many subjects in ecology, such as population dynamics (*Clark & Bjørnstad, 2004*; *Damgaard, 2012*; *Iijima, Nagaike & Honda, 2013*), metapopulation dynamics (*Harrison, Hanski & Ovaskainen, 2011*), and tree growth (*Shimatani & Kubota, 2011*; *Hiura, Go & Iijima, 2019*). The state-space model consists of two types of sub-model, the observation model and the system model; the former describes the relationships that exist between the observed data and unobserved systems, and the latter describes the processes in the unobserved latent system, such as the temporal changes of the system. Notably, this class of models has a hierarchical structure and can explicitly describe the observation processes and the latent system processes separately (*Kéry & Schaub, 2011*; *Irvine et al., 2019*). In addition, by using state-space modeling, latent states can be estimated even if there are missing observations (*Durbin & Koopman, 2012*).

The state-space model has also been used for dealing with time-series pin-point cover data (*Damgaard, 2012*). However, the cover class data treated in the present study typically have less information than pin-point cover data. Few studies have applied state-space modeling to cover class data, but if the class data were collected longitudinally, we would be able to utilize the information; i.e., the value of the latent state at a survey occasion should be similar to those at temporally adjacent occasions. In addition, if the class data were surveyed in quadrats that are arranged sequentially, we could also utilize information from the spatial autocorrelation.

In this study, a state-space model was constructed to estimate the unobserved proportion of plant cover from ordered class data using the incomplete beta function, combining information from temporal and spatial autocorrelations. This type of model would help to utilize visually determined plant cover data with temporal and spatial autocorrelation.

## METHODS

### Statistical model
#### Model basis
The beta distribution has been used to describe statistical variations in plant cover, because the distribution has a boundary from zero to one, and because it can describe various shapes (*Chen et al., 2008*; *Irvine & Rodhouse, 2010*; *Eskelson et al., 2011*; *Damgaard, 2012*; *Damgaard, 2013*; *Damgaard, 2014*; *Herpigny & Gosselin, 2015*; *Wright et al., 2017*; *Takarabe & Iijima, 2019*). In this approach, the proportion of cover $p$ $(0 < p < 1)$ is assumed to follow the beta distribution:

$$p \sim \text{Beta}(\alpha, \beta),$$

where $\alpha$ $(> 0)$ and $\beta$ $(> 0)$ are the parameters. Another parameterization using the mean of the proportion $\mu$ $(0 < \mu < 1)$ as a parameter is available (*Irvine et al., 2019*),

$$p \sim \text{Beta}(\mu\phi, (1-\mu)\phi),$$

or

$$p \sim \text{Beta}\left(\frac{\mu}{\delta} - \mu, \frac{(1-\mu)(1-\delta)}{\delta}\right),$$

where $\phi (> 0)$ and $\delta$ $(0 < \delta < 1)$ are the parameters that control the dispersion $(\phi = (1-\delta)/\delta)$. The parameter $\delta$ is also defined as the intra-quadrat correlation of the plant distribution (*Damgaard, 2012*; *Damgaard, 2013*; *Damgaard, 2014*), and it can be regarded as representing the uncertainty of the observation of the cover proportion when it is rather small (Fig. 1). In the case of $\mu = 0.5$, the distribution stays unimodal when $\delta$ is smaller than 1/3. In contrast, when $\delta$ becomes larger, the distribution tends to become bimodal (zero and one), or unimodal at zero or one (depending on $\mu$). In the parameterization set using $\delta$, the variance was given as $\delta\mu(1-\mu)$.

The probability that $p$ falls between $x_0$ and $x_1$ $(0 < x_0, x_1 < 1$, and $x_0 < x_1)$ can be described as follows:

$$\Pr(x_0 < p < x_1 | \alpha, \beta) = B(x_1, \alpha, \beta) - B(x_0, \alpha, \beta),$$

or

$$\Pr(x_0 < p < x_1 | \mu, \delta) = B\left(x_1, \frac{\mu}{\delta} - \mu, \frac{(1-\mu)(1-\delta)}{\delta}\right) - B\left(x_0, \frac{\mu}{\delta} - \mu, \frac{(1-\mu)(1-\delta)}{\delta}\right),$$

where $B(x, \alpha, \beta)$ is the cumulative density function of the beta distribution, which is identical to the regularized incomplete beta function $I_x(\alpha, \beta)$. Note that $B(0, \alpha, \beta) = 0$ and $B(1, \alpha, \beta) = 1$.

Figure 2 shows the relationship between the mean proportion ($\mu$) and the proportion of the beta distribution with a given $\mu$ and $\delta$ that is classified into each class. When the value of $\delta$ is small, the realized value of the cover class would directly correspond to the mean value. In contrast, the larger $\delta$ becomes, classes other than those corresponding to the mean tend to be chosen more frequently. For simplicity, this formulation does not explicitly account for the measurement or the detection process, though *Irvine et al. (2019)* have developed a model that explicitly incorporates a process in which the non-detection of plants and misidentification of the cover class may occur.

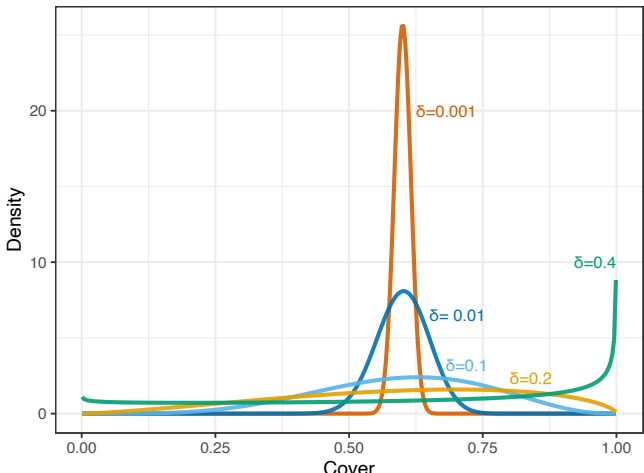

**Figure 1** Probability densities of beta distributions corresponding to the cover proportion with a fixed mean ($\mu = 0.6$) and varying the value of $\delta$ (0.001, 0.01, 0.1, 0.2, and 0.4).

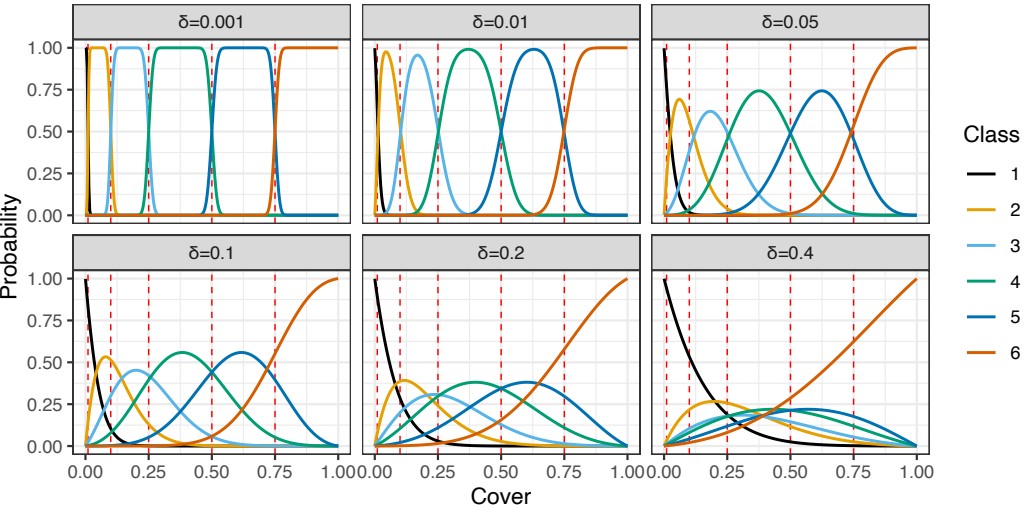

**Figure 2** Relationships between the mean plant cover proportion ($\mu$) and the proportion of the realized values for given $\mu$ and $\delta$ that are classified into each class, with varying values of $\delta$ (0.001, 0.01, 0.05, 0.1, 0.2, and 0.4). Red dashed lines show cut points (inner boundaries of the classes).

### State-space model

*Observation model.* Assume that surveys on plant cover were conducted $N_T$ times in $N_Q$ quadrats. In the present study, quadrats were assumed to be arranged on a line.

The cover class, $Y$, was defined as six classes corresponding with the proportion of plant cover as follows:

$$Y = \begin{cases} 1 & \text{if } 0 \leq \text{cover} \leq 0.01, \\ 2 & \text{if } 0.01 < \text{cover} \leq 0.1, \\ 3 & \text{if } 0.1 < \text{cover} \leq 0.25, \\ 4 & \text{if } 0.25 < \text{cover} \leq 0.5, \\ 5 & \text{if } 0.5 < \text{cover} \leq 0.75, \\ 6 & \text{if } 0.75 < \text{cover} \leq 1. \end{cases}$$

In reality, $Y$ would be typically determined with visual measurements.

In this study, estimating the cover proportion of a particular species was the primary purpose rather than the presence/absence of the species. Thus, for simplicity, the model did not distinguish the absence of the species (or more precisely, the absence of the detection of the species) from the smallest plant cover class. When the plant species richness of the area is the study purpose, both states should be modeled separately. In those cases, incorporating zero-inflation (*Herpigny & Gosselin, 2015*) and the correction of false-negative errors (*Chen et al., 2009*; *Chen et al., 2013*) into the model is required. *Irvine et al. (2019)* have explicitly modeled this observation process to estimate latent cover proportions and detection errors.

The relationship between the observation $Y_{t,q}$, the cover class at time $t \in \{1, 2, \ldots, N_T\}$ and in quadrat $q \in \{1, 2, \ldots, N_Q\}$, and $\mu_{t,q}$, the mean proportion of plant cover at time $t$ in quadrat $q$, was defined after *Irvine et al. (2019)* as follows:

$$\Pr(Y_{t,q} = Y | \mu_{t,q}, \delta) = F_{\text{BETA}}(d_Y, \mu_{t,q}, \delta) - F_{\text{BETA}}(d_{Y-1}, \mu_{t,q}, \delta)$$

where $F_{\text{BETA}}(d_Y, \mu_{t,q}, \delta)$ denotes the regularized incomplete beta function $B\left(d_Y, \frac{\mu_{t,q}}{\delta} - \mu_{t,q}, \frac{(1 - \mu_{t,q})(1 - \delta)}{\delta}\right)$ and $d_Y$ denotes the cut points. In this study, $d_Y$ were defined as $\{0.01, 0.1, 0.25, 0.5, 0.75\}$ for $Y \in \{1, 2, \ldots, 5\}$, corresponding to the definition of $Y$. In addition, $d_0$ and $d_6$ were defined to be 0 and 1, respectively, so that $F_{\text{BETA}}(d_0, \mu, \delta) = 0$ and $F_{\text{BETA}}(d_6, \mu, \delta) = 1$.

The mean proportion of plant cover $\mu_{t,q}$ was defined by incorporating the latent state $\theta_t$ at time $t \in \{1, 2, \ldots, N_T\}$,

$$\mu_{t,q} = \text{logit}^{-1}(\theta_t + r_{t,q}),$$

where $r_{t,q}$ denotes the spatial random effect incorporating spatial autocorrelation.

*System model.* The latent state $\theta_t$ at time $t$ denotes the states related to the mean proportion of plant cover, and the expected mean proportion of plant cover at time $t$ for the overall plots, $\phi_t$, is given as

$$\phi_t = \text{logit}^{-1}(\theta_t).$$

The transition of the latent state $\theta_t$ was defined using second-order differences with normal error as follows:

$$\theta_t - \theta_{t-1} = \theta_{t-1} - \theta_{t-2} + \epsilon_T \quad \text{for } t \in \{3, 4, \ldots, N_T\}$$

$$\epsilon_T \sim \text{Normal}(0, \sigma_T^2),$$

therefore,

$$\theta_t | \theta_{t-1}, \theta_{t-2}, \sigma_T \sim \text{Normal}(2\theta_{t-1} - \theta_{t-2}, \sigma_T^2),$$

where $\sigma_T$ denotes the standard deviations. This formulation is equivalent to the second-order autoregressive (AR(2)) model, $\theta_t = c + a_1\theta_{t-1} + a_2\theta_{t-2} + \epsilon_T$ with $c = 0$, $a_1 = 2$, $a_2 = -1$.

Priors of the latent states at time $t \in \{1, 2\}$ were defined as weakly informative (*Gelman, Simpson & Betancourt, 2017*) so that they would be effective for model identifiability but not strongly restrict the range of the posterior distributions; they should be wide enough for the logit-scaled parameters (in case $\theta = 5$, $\phi$ is 0.99),

$$\theta_t \sim \text{Normal}(0, 2.5^2) \quad \text{for} \quad t \in \{1, 2\}.$$

The spatial random effect $r_{t,q}$ at time $t$ of quadrat $q$ was defined as follows:

$$r_{t,q} - r_{t,q-1} | \sigma_R \sim \text{Normal}(0, \sigma_R^2) \quad \text{for} \quad q \in \{2, 3, \ldots, N_Q\}$$
$$r_{t,1} \sim \text{Normal}(0, 2.5^2),$$

where $\sigma_R$ denotes the standard deviation among the spatial random effects. The value of the random effect $r_{t,q}$ was assumed to be affected by those of the adjacent quadrats. This formulation was equivalent to a process model of a state-space model with a first-order difference in the state changes. Then, the values were updated so that their sum should be zero for each survey time to avoid affecting the overall intercept and the identifiability of the model.

$$r_{t,q} \leftarrow r_{t,q} - \frac{1}{N_Q}\sum_{j=1}^{N_Q} r_{t,j}.$$

Priors for standard deviation parameters $\sigma_R$ and $\sigma_T$ were defined as weakly informative for the same reason as for $\theta_1$ and $\theta_2$, as follows:

$$\sigma \sim \text{HalfNormal}(0, 2.5^2).$$

## Application to simulated data
### Generation of simulated data
Assume that there were $N_T = 10$ quadrats that settled sequentially, and plant cover classes were surveyed for $N_Q = 15$ times in each quadrat. A simulated dataset was generated according to this assumption. In the simulated data, the parameter $\theta_t$, which denotes the latent state at time $t$, was generated following the relationship below:

$$\theta_1 = -6$$
$$\theta_t \sim \text{Normal}(\theta_{t-1} + 0.3, 0.5^2) \quad \text{for} \quad t \in \{2, 3, \ldots, N_T\}.$$

The latent state $\theta_t (t \in \{2, 3, \ldots, N_T\})$ was randomly generated following the above normal distribution. Note that the first-order difference was used in this data generation, for

simplicity, while the second-order difference was adopted in the model defined above. The spatial random effects $r_q(1 \in \{2, 3, \ldots, N_Q\})$ were also generated randomly, with the assumption of following the above normal distribution. In this simulation, the spatial random effects were assumed to be invariant through time.

$$r_1 = 0$$

$$r_q \sim \mathrm{Normal}(r_{q-1}, 0.5^2).$$

Proportions of plant cover $p$ were generated according to the model defined in the previous subsection:

$$p_{t,q} = \mathrm{logit}^{-1}(\theta_t + r_q).$$

Then, the plant cover classes were generated with an uncertainty $\delta$. In this simulation, the value of $\delta$ was set to 0.05. The parameter $\delta$ can be defined as the intra-quadrat correlation of plant distribution in the pin-point cover data (*Damgaard, 2012*), but, in this simulated data, it was decoupled from the parameter $\sigma_Q$, which control the similarity between neighboring quadrats.

The cover classes adopted in this simulated data were as follows: 1 (for proportion 0–0.01, including 0), 2 (0.01–0.1), 3 (0.1–0.25), 4 (0.25–0.5), 5 (0.5–0.75), and 6 (0.75–1). The generated data are shown in Fig. 3. The data generation code is available at the GitHub repository (https://github.com/ito4303/ssmcover).

### Fitting to the model

The generated data were fit to the Bayesian state-space model defined in the above subsection, and the posterior distributions of each parameter were estimated using the Markov chain Monte Carlo (MCMC) method. The model was implemented using Stan version 2.21.0 (*Carpenter et al., 2017*) with the re-parameterization of the model for the stability and efficiency of the Hamiltonian Monte Carlo algorithm, which was adopted in the Stan software. The Stan model code is also available at the GitHub repository. Posterior samples were drawn from 1,000 iterations after 1,000 warm-up (burn-in) iterations from each of 4 chains, and the posterior distributions of the parameters were estimated. Then, posterior predictive checks were conducted to evaluate the fitting to the model using the 'bayesplot' package (*Gabry et al., 2019*) in the statistical software R version 3.6.2 (*R Core Team, 2019*). In the posterior predictive check, the data drawn from the posterior predictive distribution that was calculated under the model were compared to the observed data using the rootogram (*Kleiber & Zeileis, 2016*), which plots the expected values under the model on the histogram of the observed data with a square-root-scaled $Y$-axis. If there are considerable discrepancies between them, it indicates that the model poorly explains the observed data.

## Application to real data

Real data to be fitted to the model were taken from long-term vegetation monitoring following a catastrophic windthrow (*Itô et al., 2018*). The data were collected during the period of 1957 to 2017 in the headwater region of the Ishikari River, Hokkaido, northern

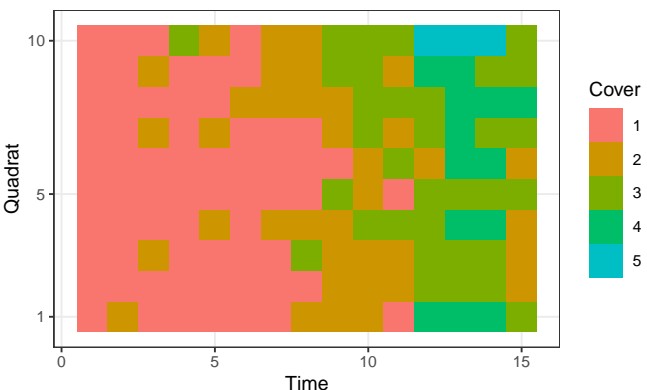

**Figure 3 Simulated data that were generated for 10 sequential quadrats and 15 survey times.** Classes denoted as follows, 1: 0–0.01 (including 0), 2: 0.01–0.1, 3: 0.1–0.25, 4: 0.25–0.5, 5: 0.5–0.75. Class 6 (0.75–1) was absent from these data.

Japan. Six plots (No. 27, 30, 34, 35, 46 and 54) were settled in the region in 1955. Quadrats sized two meters × two meters were settled sequentially, and the number was 15–25 for each plot. The visually determined cover classes were recorded for species that occurred in each quadrat.

The classes used in the surveys were as follows: + (proportion: 0–0.01, excluding zero), 1 (0.01–0.1), 2 (0.1–0.25), 3 (0.25–0.5), 4 (0.5–0.75), and 5 (0.75–1). Species that were not detected (i.e., the cover was 0) did not appear in the dataset. However, in the analysis, the notation was changed to be identical to the simulated data shown above for the sake of simplicity in numerical treatments so that the absence (more precisely, non-detection) was added to class 1 (0–0.01, including zero). The dataset is also available at the GitHub repository since it was published under the license CC BY 4.0.

From this dataset, cover classes of a species of dwarf bamboo, *Sasa senanensis*, in the shrub layer of six plots were used as the real data to be fit to the Bayesian state-space model. The data of the species had a wide variation in the cover class measurements and were suitable for model evaluation. The plots had 19–25 quadrats, and the survey was conducted 20 times (in 1957–1968, 1972, 1976, 1980, 1984, 1988, 2002, 2009, and 2017). Though the measurements were not conducted in all years during the period (1957–2017), the latent state could be estimated using the state-space model (*Durbin & Koopman, 2012*). Figure 4 shows the changes in cover classes.

The posterior distributions were estimated using the MCMC method. Stan was also used for the estimation, and the posterior samples were drawn from 2,000 iterations after 2,000 warm-up (burn-in) iterations from each of 4 chains. Then, the posterior predictive checks were conducted using the 'bayesplot' package.
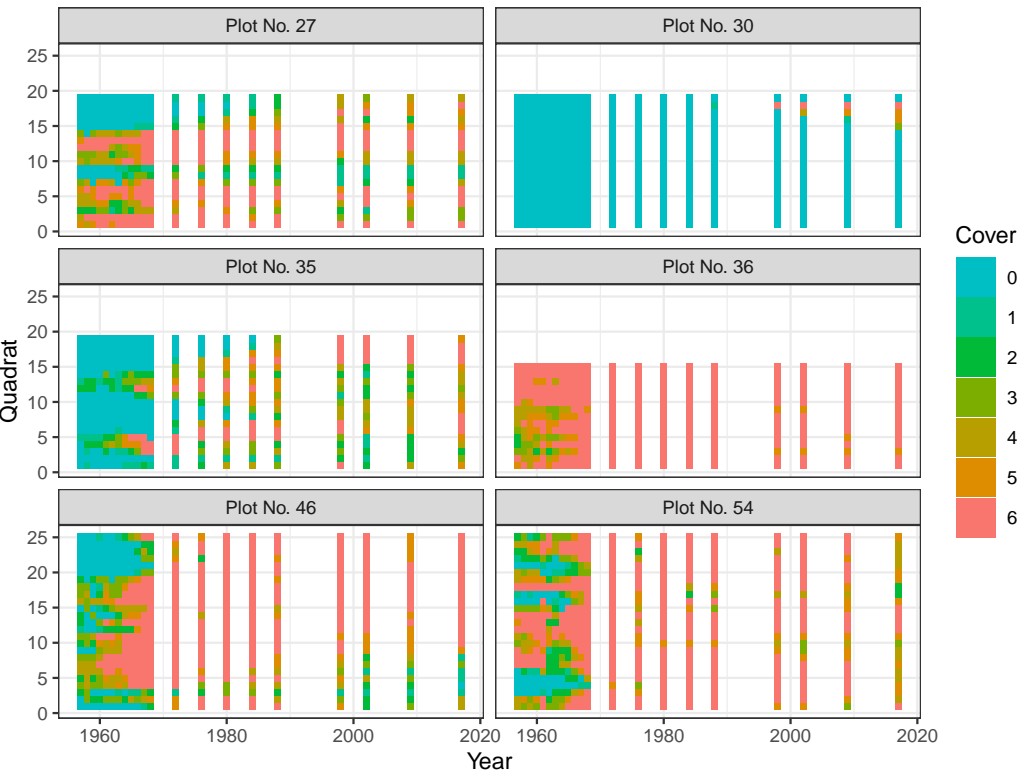

**Figure 4** **Changes in plant cover classes of *Sasa senanensis* following a catastrophic windthrow for six plots in Hokkaido, Japan.** Classes are denoted as follows, 0: 0, 1: 0–0.01 (excluding 0), 2: 0.01–0.1, 3: 0.1–0.25, 4: 0.25–0.5, 5: 0.5–0.75, 6 0.75–1. In the model fitting, class 0 was incorporated into class 1 because the model did not distinguish between these classes.

## RESULTS

### Simulated data

Gelman–Rubin statistics, $\hat{R}$, (*Gelman & Rubin, 1992*; *Brooks & Gelman, 1998*) were smaller than 1.1 for all the parameters, suggesting that the Markov chain successfully converged to the stationary distribution.

Table 1 shows the summary of the posteriors for the parameters $\delta$, $\sigma_T$, and $\sigma_R$. The posterior means (and 95% credible intervals) of these parameters were estimated as 0.06 (0.03–0.09) for $\delta$, 0.34 (0.08–0.62) for $\sigma_R$, and 0.75 (0.32–1.43) for $\sigma_T$ (Table 1). The values used for the data generation were 0.05, 0.5, and 0.5, respectively.

Figure 5 shows the overall cover proportion ($\phi = \text{logit}^{-1}(\theta)$) calculated from the posterior median (the red line) and the 95% credible intervals (the red region) as well as the cover classes in the simulated data (black dots) and the cover proportion averaged for each time (black curve). The posterior predictive check showed no conflicts between the observed value and the predicted distribution for each time. Figure 6 shows the result at time 15.
**Table 1** Summary of the posteriors of the parameters $\delta$, $\sigma_R$, and $\sigma_T$ for the simulated data.

| Parameter | Mean | Percentile | | | $\hat{R}$ |
|---|---|---|---|---|---|
| | | 2.5% | 50% | 97.5% | |
| $\delta$ | 0.06 | 0.03 | 0.06 | 0.09 | 1.00 |
| $\sigma_R$ | 0.34 | 0.08 | 0.34 | 0.62 | 1.01 |
| $\sigma_T$ | 0.75 | 0.32 | 0.71 | 1.43 | 1.00 |

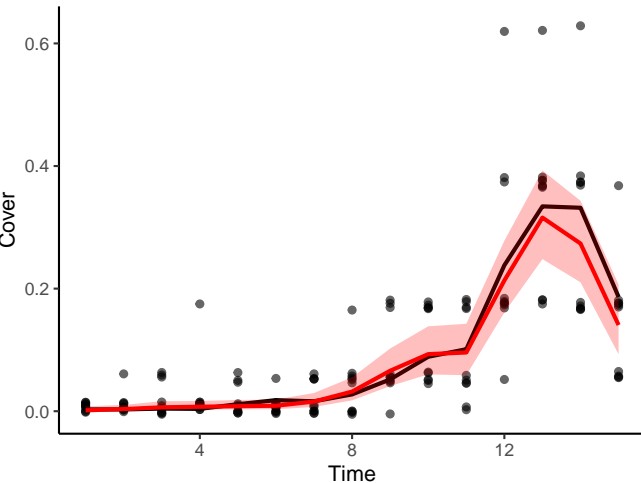

**Figure 5** **Estimated values of the cover proportion with simulated cover class data.** Black curve: mean cover proportions that were used to generate cover class simulated data (averaged within each time). Black dots: cover classes in the simulated data (dots are on the medians of the classes and are jittered vertically). Red curve: estimated overall cover proportion without spatial variations. Red region: 95% credible intervals of the estimated cover proportion.

## Real data

$\hat{R}$ values were smaller than 1.1 for all parameters, and the Markov chain seemed to converge to the stationary distribution.

Table 2 shows a summary of the posteriors of the parameters $\delta$, $\sigma_T$, and $\sigma_R$ for each plot. The posterior means of the uncertainty or intra-plot correlation parameter $\delta$ were estimated as 0.12–0.31, those of the spatial variability parameter $\sigma_R$ were 0.67–3.06, and those of the temporal variability parameter $\sigma_T$ were 0.04–0.63 (Table 2).

Figure 7 shows the estimated overall cover proportions for each year for each plot. Figure 8 shows the results of the posterior predictive checks in 2017 for each plot. The posterior predictive checks showed little conflict between the observed values and the predicted distribution for each year except for a few of the predicted values such as that of class 5 of plot 30 and that of class 4 of plot 54.

## DISCUSSION

For the simulated data, the posterior means (and medians) for the three major parameters did not differ much from the values that were used in the data generation (Table 1). The

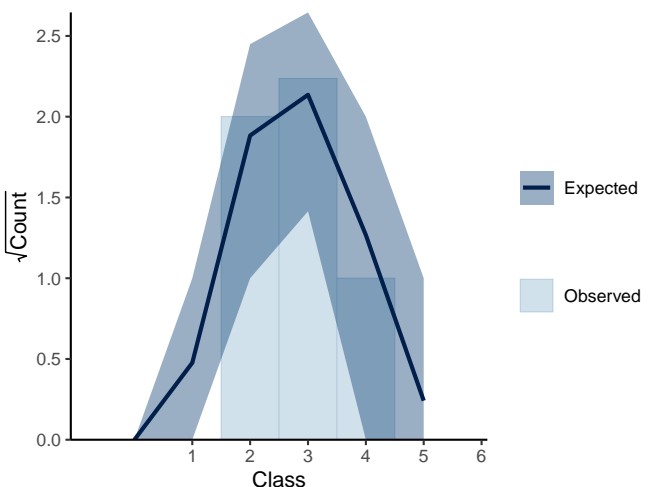

**Figure 6** **Rootogram showing the posterior predictive check for the Bayesian inference of the simulated data.** The bars show the measurements for each cover class observed at time 15; the curve shows the expected value of the posterior predictive distribution, and the dark region shows the 90% credible intervals at time 15.

**Table 2** **Summary of the posteriors of the parameters $\delta$, $\sigma_R$, and $\sigma_T$ for the real data.**

| Plot No. | Parameter | Mean | Percentile | | | $\hat{R}$ |
|---|---|---|---|---|---|---|
| | | | 2.5% | 50% | 97.5% | |
| 27 | $\delta$ | 0.15 | 0.06 | 0.15 | 0.27 | 1.01 |
| | $\sigma_R$ | 3.06 | 2.37 | 3.06 | 3.78 | 1.01 |
| | $\sigma_T$ | 0.04 | 0.01 | 0.04 | 0.11 | 1.00 |
| 30 | $\delta$ | 0.78 | 0.58 | 0.79 | 0.92 | 1.01 |
| | $\sigma_R$ | 1.06 | 0.40 | 1.00 | 2.02 | 1.01 |
| | $\sigma_T$ | 0.25 | 0.03 | 0.20 | 0.81 | 1.00 |
| 35 | $\delta$ | 0.12 | 0.04 | 0.11 | 0.24 | 1.01 |
| | $\sigma_R$ | 2.60 | 1.93 | 2.60 | 3.24 | 1.01 |
| | $\sigma_T$ | 0.06 | 0.02 | 0.05 | 0.13 | 1.00 |
| 36 | $\delta$ | 0.24 | 0.16 | 0.24 | 0.32 | 1.01 |
| | $\sigma_R$ | 0.67 | 0.40 | 0.66 | 1.05 | 1.01 |
| | $\sigma_T$ | 0.16 | 0.05 | 0.14 | 0.38 | 1.00 |
| 46 | $\delta$ | 0.24 | 0.16 | 0.24 | 0.33 | 1.01 |
| | $\sigma_R$ | 1.60 | 1.19 | 1.59 | 2.08 | 1.01 |
| | $\sigma_T$ | 0.27 | 0.10 | 0.25 | 0.54 | 1.00 |
| 54 | $\delta$ | 0.31 | 0.14 | 0.32 | 0.48 | 1.01 |
| | $\sigma_R$ | 1.73 | 0.80 | 1.70 | 2.78 | 1.01 |
| | $\sigma_T$ | 0.63 | 0.26 | 0.60 | 1.18 | 1.00 |

estimated curve of the plant cover proportion was similar to that which generated the simulated data (Fig. 5). The result of the posterior predictive check (Fig. 6) also suggests little discrepancy between the fitted model and the simulated dataset. The differences

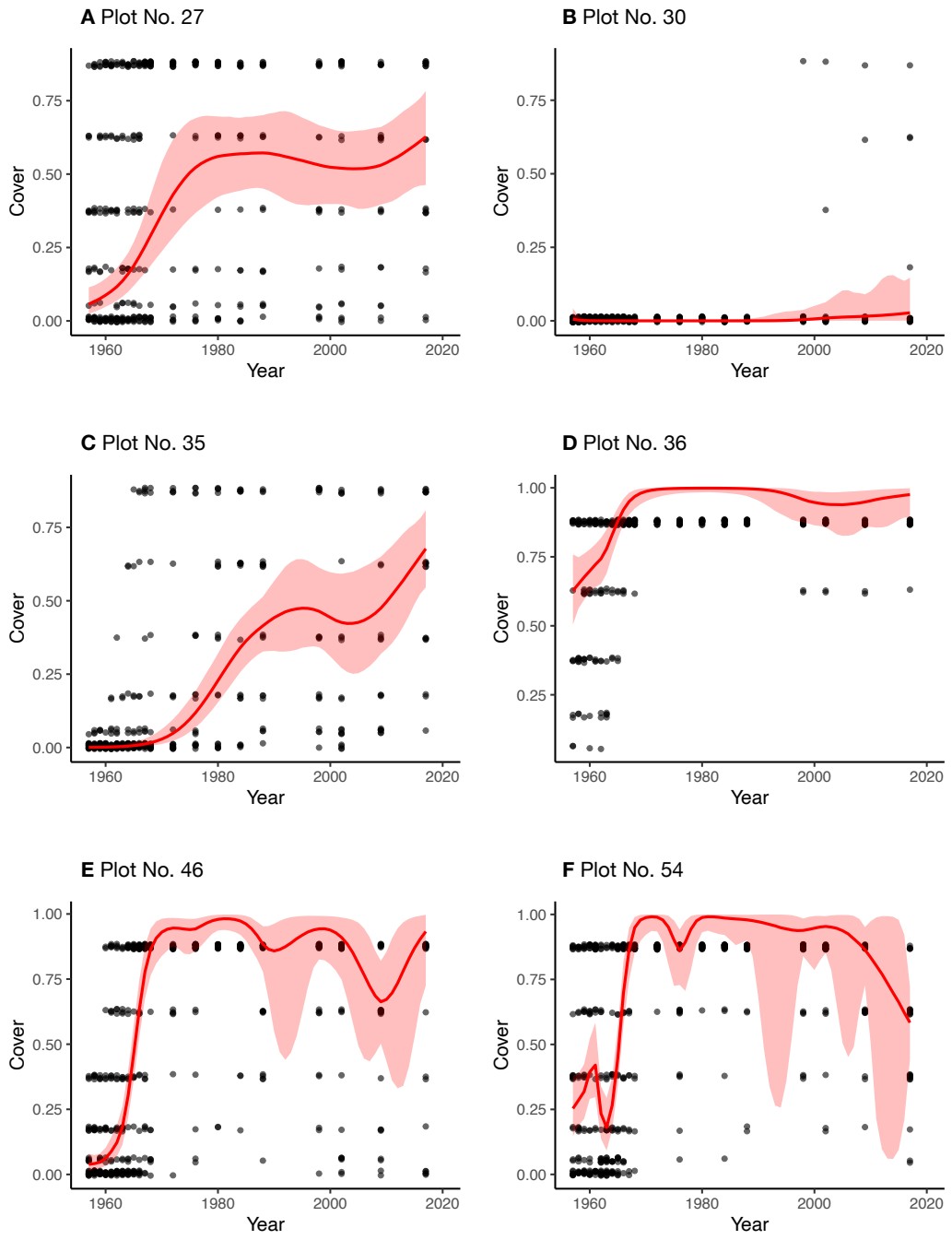

**Figure 7** **Real data and estimated values of the plant cover proportion for each plot (A–F).** Black dots: observed cover classes in the real data (dots are on the medians of the classes and are jittered vertically). Red curves: estimated overall cover proportion without spatial variations. Red regions: 95% credible intervals of the estimated cover proportion.

between the posterior means and the original values in parameters $\sigma_R$ and $\sigma_T$ may be at least partially due to variations in the randomly generated data. However, the slightly smaller value of $\sigma_R$ may be attributable to small variations of cover classes among quadrats
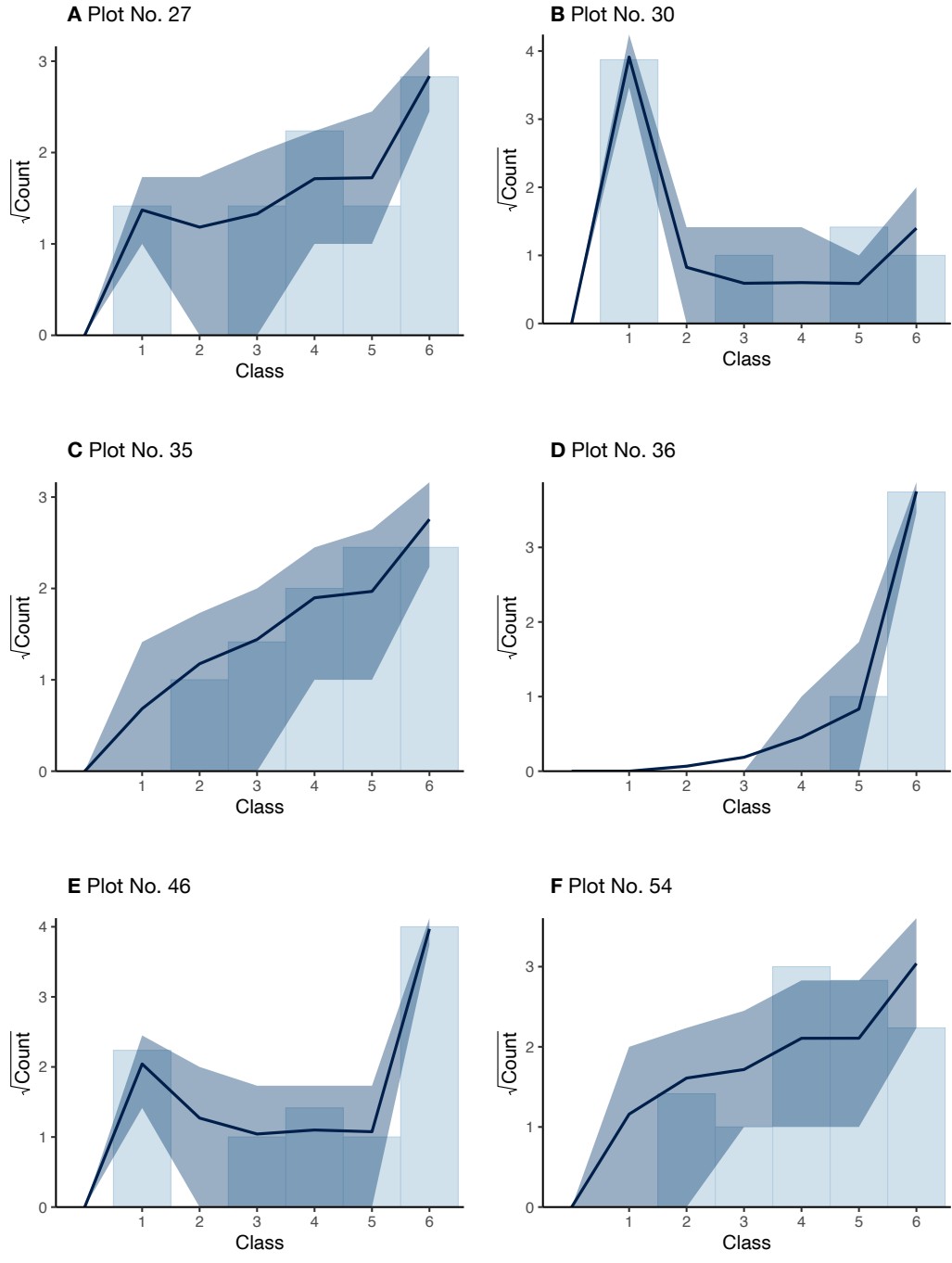

**Figure 8** Rootogram showing the posterior predictive check for the Bayesian inference of the real data for each plot (A–F). The bars show the measurements for each cover class observed in the year 2017; the curves show the expected values of the posterior predictive distribution, and the dark regions show the 90% credible intervals.

in the first several surveys (Fig. 3). Over the period, the small value of $\theta_t$ overwhelmed the value of $r_t$. In addition, the small variations in the period would affect the narrow credible intervals of the posteriors (Fig. 5). Also, the assumption of the second-order differences in the system model rather than the first-order differences used in the data generation may have affected the difference in $\sigma_T$.

For the real data, the estimated curves of the overall plant cover proportion seem to be reasonable when comparing the measured cover classes for each plot (Fig. 7). Most of the posterior predictive checks (Fig. 8) also suggest little conflict between the model and the real datasets. Owing to the state-space modeling, the overall proportions could be estimated including the unobserved years, but the credible intervals were considerably wider for some plots (plots 46 and 54), especially in later years without surveys. This is mainly because of lack of information due to the sparse survey intervals.

The posterior mean of $\sigma_R$ was largest (3.06) in plot 27 (Table 2), although the range did not seem strongly affected by the prior (HalfNormal(0, 2.5)). This is likely due to the somewhat more considerable variation in the measurements among the adjacent quadrats (Fig. 4). On the other hand, the value was the smallest in plot 36 (Table 2), reflecting the small variations among quadrats.

The estimated posterior means of $\delta$ ranged from 0.12 to 0.78 (Table 2), and these values suggest that the uncertainty or intra-plot correlation was somewhat large (Fig. 2). In particular, the value was largest in plot 30 (0.78), where most of the observed values were zero (Fig. 8). This is reasonable because the intra-quadrat correlation of the plant cover should be large when most values are zero. In this situation, it may be adequate to interpret the value of $\delta$ as the intra-quadrat correlation rather than uncertainty. On the other hand, the posterior mean of $\sigma_R$, the scale parameter of spatial variation in the logit-scale, was not so small (1.06) compared to the simulated data (0.34). This may be because the overall mean of the cover proportion in the logit-scale was so small that $\sigma_R$ weakly affected the likelihood in this case.

If the intra- and inter-quadrat distributions are related, the inter-quadrat variation within a year also may correlate with the intra-quadrat correlation. An integrated evaluation of the intra- and inter-quadrat correlation may enable us to evaluate the spatial distribution pattern of a target plant at various scales.

The state-space modeling seems to have successfully estimated the changes in the latent states in the years that the surveys were not conducted. These results suggest that the present model is applicable to this type of plant cover class data.

Though the model proposed in this study is rather simple, more elaborate models can be constructed. For example, the one-dimensional structure of the present model can be expanded to two dimensions. To incorporate a two-dimensional spatial autocorrelation, conditional autoregressive (CAR) models can be utilized, and they are available in Stan (*Joseph, 2016*; *Morris et al., 2019*) Another possible expansion is to incorporate zero-inflation. *Herpigny & Gosselin (2015)* and *Irvine et al. (2019)* have already provided modeling of plant cover classes with zero-inflation. When incorporating this, false-negative errors should be considered (*Chen et al., 2009*; *Chen et al., 2013*; *Irvine et al., 2019*). In

addition, misclassification of the cover classes should be explicitly incorporated (*Irvine et al., 2019*).

## CONCLUSION

State-space modeling for plant cover class data can successfully estimate the unobserved cover proportion by utilizing spatial and temporal autocorrelations that are contained within the data. The present model can be applicable to similar types of plant cover class data, and then can be expanded to deal with two-dimensional field data, or to incorporate zero-inflation and misclassification of the cover classes.

## ACKNOWLEDGEMENTS

I thank Dr. H Iijima (Forestry and Forest Products Research Institute, Japan) for useful comments on a previous version of this paper. I also thank two reviewers (Prof. C Damgaard and Dr. KM Irvine) for their valuable suggestions to improve this paper.

### Funding
The author received no funding for this work.

### Competing Interests
The author declares there are no competing interests.

### Author Contributions
- Hiroki Itô conceived and designed the experiments, performed the experiments, analyzed the data, prepared figures and/or tables, authored or reviewed drafts of the paper, and approved the final draft.

### Data Availability
All program codes and data are available at GitHub:

https://github.com/ito4303/ssmcover.

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
