# Peer review of "State-space modeling of the dynamics of temporal plant cover using visually determined class data"

_PeerJ, doi:10.7717/peerj.9383_

## Round 0.1 · original submission · Major Revisions

The two reviewers - experts of plant cover modelling - saw value in your paper, but they made a number of important comments regarding the model formulation and how your model and its implementation in stan are related to current literature.

·

Basic reporting

NA

Experimental design

NA

Validity of the findings

NA

Additional comments

It is an important subject. However, I have some reservations regarding the model and generally the English could be improved.

The formulas are not numbered so in the following line 113+ means the formula after 113.

Line 113+. I do not understand the time-series model. Why not an AR2 model (that should be discussed) and what happens if years are missing as is the case for the empirical data set.

Line 117+. The spatial variation among quadrats are here modelled by a random variable. But this variation is also modelled by delta. I suggest you save the random variable for among-plot variation and test if delta are adequate to model the within-plot variation. In the current model, the parameter delta cannot be interpreted.

Line 122+. Why not use a strong prior instead (as in line 117+), please explain.

Line 169. Only one plot is modelled. It would be more convincing if all plots were modelled (see also comment to line 117+).

Figure 4. Explain that class 0 and 1 was modelled as 1 class (if this is the case).

Figure 5. The mentioned red and black curves are missing.

Figure 6. I am not familiar with a “rootogram”, please explain

·

Basic reporting

no comment

Experimental design

no comment

Validity of the findings

no comment

Additional comments

please see my uploaded comments on the pdf of the article.
I feel sort of awkward basically saying "you should cite my work." But I hope providing those connections in the literature are helpful for all of us. My main confusion was around the highlighted section and the intepretation of Figure 2, but refer to my commented pdf.

---

## Round 0.2 · accepted · Accept

Both reviewers found your revision well done, and I concur with their assessment. Nice paper!

·

Basic reporting

na

Experimental design

na

Validity of the findings

na

Additional comments

I find that the author adequately has responded to my comments

·

Basic reporting

no comment

Experimental design

no comment

Validity of the findings

no comment

Additional comments

I appreciate the author's work to respond to my previous comments.
I found this work interesting and a contribution to the world of modeling plant cover data.
K.M. Irvine